# Effects of LED Applications on Dahlia (*Dahlia* sp.) Seedling Quality

**DOI:** 10.3390/plants14152319

**Published:** 2025-07-27

**Authors:** Gamze Gündoğdu, Murat Zencirkıran, Ümran Ertürk

**Affiliations:** 1Department of Hortıculture, Faculty of Agriculture, Bursa Uludağ Unıversıty, Bursa 16059, Türkiye; umrane@uludag.edu.tr; 2Landscape Architecture Department, Faculty of Agriculture, Bursa Uludağ Unıversıty, Bursa 16059, Türkiye; mzencirkiran@uludag.edu.tr

**Keywords:** diversity, LED application, light intensity, quality

## Abstract

This study aimed to determine the effects of LED applications and application periods on seedling development. To this end, four different LED applications (blue 100%, red 100%, green 100%, and full-spectrum 100% (control)) were applied to different star flower varieties (Figaro Violet shades—flower color: purple, Figaro Orange shades—flower color: orange, Figaro White shades—flower color: white, and Figaro Red shades—flower color: red) for 15 and 30 days. These applications were repeated over two years (two vegetation periods). The results revealed that the red-flowered and white-flowered varieties exhibited higher values in terms of root length, root number, stem diameter, 2nd and 4th leaf petiole length, 2nd and 4th leaf width, and leaf number under full-spectrum and red LED applications. We also observed that red LED application for 30 days is suitable for seedling height development in the Figaro Orange shades variety. Conversely, the results showed that the effects of LED application durations on root length and stem diameter did not show a statistically significant difference, while the 15-day application yielded the best results for root number. In the Figaro Red shades and Figaro White shades varieties, the use of red LED applications for 30 days yielded results similar to those of full-spectrum applications, indicating that both applications can be used for seedling cultivation.

## 1. Introduction

As an energy source during photosynthesis and a trigger for a series of developmental events [1], light is essential for plant growth and development [2]. A plant’s photomorphogenetic development can be altered by the properties of light [3]. Photomorphogenesis is reported to encompass various stages of plant growth and development directly controlled by light, such as seed germination, seedling development, and the transition from the vegetative phase to the generative phase [1]. Plants use photosynthetically active radiation (PAR) from light with a wavelength of 400–700 nm [4]. Photosynthesis is the process broadly defined as the process of energy conversion from light or photosynthetically active radiation (PAR) into chemical and heat energy, measured in joules. The conversion from joules to moles is determined after the wavelength is specified [5]. The quality or spectral distribution of light on the plant canopy significantly differs from that within it. Green light passes through the plant canopy more easily than blue and red light, because the transmittance of green leaves is 30%, reflectance is 20%, and absorption rates are approximately 50%. Blue and red light are mostly absorbed by the uppermost layer of the plant canopy [6]. Blue and red LED lights significantly affect plant growth and yield, while the duration of lighting is reported to affect flowering [7]. Blue LEDs play an important role in chloroplast development, chlorophyll formation and stomatal opening [8,9,10]. A combination of blue and red LEDs promotes seedling growth and development [11]. LED combinations are reported to have positive effects on yield and quality [12].

Numerous previous studies documented different effects of light color on plants [2,3,4,5,6,7,8,9,10,11,12,13,14]. For this purpose, artificial light sources are widely used, especially in greenhouse cultivation, indoor ornamental plant cultivation, and tissue culture studies. Recent technological advances have led to the widespread use of low-energy, high-efficiency LEDs [15]. As a result, in recent years, LEDs have been developed to provide an appropriate light spectrum (quality and duration) in a controlled environment [16,17].

The photoperiod refers to the length of time a plant is exposed to light [18]. LED light cycles considerably affect plant physiology, influencing flowering and seed germination. Adjusting the photoperiod can modulate plant growth rates from seedling to maturity, potentially shortening vegetable growth cycles [19].

Since the effects of LED lighting vary for each plant genotype [20], it is of great importance to increase research on a species-by-species basis to fully determine the effectiveness of specific wavelengths on plant growth.

In this context, the present study aims to determine the effects of different LED wavelengths and application durations on seedling quality in a *Dahlia* ssp. (Star Flower), a perennial herbaceous and tuberous flower classified as a cut flower or garden flower.

## 2. Materials and Methods

### 2.1. Materials

Four varieties of Dahlia seeds were used as study materials: (i) Figaro Violet shades (flower color: purple); (ii) Figaro Orange shades (flower color: orange); (iii) Figaro White shades (flower color: white); and (iv) Figaro Red shades (flower color: red). The seeds were obtained from TASACO (Türkiye).

### 2.2. Methods

In the first year, the seeds were machinesowed into 128-cell trays measuring 53 × 27.5 cm and then filled with germination peat (pH: 5.8–6.2) in March. In the second year, seeds were again obtained and machinesowed in March. Next, the trays were stored without light under controlled conditions with an average temperature of 18–20 °C and 70–80% humidity to facilitate germination. Full spectrum (control) light applications were initiated after 50% germination had occurred to prevent plants from becoming etiolated due to insufficient light conditions, with the lights positioned 15 cm above the tray level. Full spectrum light was used daily for 14 h. Since these were long-day plants, 14 h LED use was preferred. LED applications were initiated after the emergence of the second true leaves in 50% of the seedlings. To this end, LEDs were used at four different wavelengths, measured as photosynthetic photon flux density (PPFD). The wavelengths were blue (B) at 455 ±5 nm 100% (PAR 6.163 mW/cm^2^, PPFD 237.661 µmol/m^2^/s, and PPFD_B 231.921 µmol/m^2^/s), red (R) at 660 ±5 nm 100% (PAR 0.046 mW/cm^2^, PPFD 2.408 µmol/m^2^/s, and PPFD_R 2.262 µmol/m^2^/s), green (G) at 515 ± 5 nm 100% (PAR 0.517 mW/cm^2^, PPFD 23.021 µmol/m^2^/s, and PPFD_G 21.177 µmol/m^2^/s) and full-spectrum light (control) with a color temperature of 3000 Kelvin (PAR 3.740 mW/cm^2^ and PPFD 175.078 µmol/m^2^/s). The LED light was used on the plants for two periods of either 15 or 30 days. The plants received light for 14 h per day with the remaining 10 h in darkness. Spray irrigation was repeated every 2 days and liquid fertilizer (NPK: 15-5-30+TE, 5 cc/10 lt) was applied every 2 weeks. No plant protection product was used. The seedlings were transferred to an acclimating greenhouse with an average temperature of 10–26 °C (night/day) and 50–85% humidity. After being kept under these conditions for 10 days, the seedlings were assessed (Figure 1).

### 2.3. Statistical Analysis

The study was conducted according to a “randomized block design” with three replicates and six seedlings in each replicate, using a total of 1152 seedlings in the first and second years. The obtained data were first subjected to analysis of variance using the Jump 5.0.1 statistical program. Then, the groups formed based on the differences (least significant difference) were determined using the LSD test.

## 3. Results

Table 1, Table 2, Table 3 and Table 4 and Figure 2, Figure 3, Figure 4 and Figure 5 summarize the results of the statistical evaluations of the average data for different LED applications carried out for different periods in the first and second years.

The results of comparing the data for the first and second years revealed that the average seedling height was 12.91 cm, root length was 4.73 cm, root number was 4.06, stem diameter was 2.66 mm, number of leaves was 7.62, tuber formation was 0.11 (Figure 4).

The results of comparing the data for the first and second years revealed that the average second leaf stem length was 1.99 cm, second leaf width was 2.03 cm, second leaf length was 4.42 cm, fourth leaf stem length was 1.48 cm, fourth leaf width was 1.89 cm, fourth leaf length was 3.79 cm (Figure 4).

The results of comparing the data for the first and second years revealed that the highest seedling height was 13.80 cm in the orange-flowered variety, while the lowest seedling heights were 12.59 cm and 12.17 cm in the purple- and red-flowered varieties, respectively (see Table 1). The analysis of the root length data for the first and second years, as well as root number data, revealed that, the red-flowered variety had the highest root length at 4.9 cm and root number at 4.28. An examination of the data for the first and second years in terms of leaf count revealed an increase in leaf count in the white- and red-flowered varieties, with 7.75 and 7.97 leaves, respectively, while the purple- and orange-flowered varieties yielded 7.37 and 7.40 leaves, respectively. Furthermore, with regard to the tuber formation data for the first and second years, we observed that the red-flowered variety formed the fewest tubers among the varieties (see Table 1).

The results of comparing the data for the first and second years revealed that the highest seedling height was 15.50 cm in the red LED application, while the lowest seedling heights were 10.82 cm and 10.52 cm in the green and blue LED applications, respectively (see Table 2, Figure 2 and Figure 3). The analysis of the root length data for the first and second years revealed that the green LED application had the lowest root length at 3.9 cm. In terms of root number, the full-spectrum LED application yielded the best result (4.43 roots), while the green LED application yielded the lowest value (3.58 roots). When examining the data for the first and second years in terms of stem diameter, no statistical difference was observed between the varieties, while the full-spectrum LED application had the highest value at 2.8 mm. In the red and blue LED applications, the values amounted to 2.67 mm and 2.73 mm, respectively, while the green LED application yielded the thinnest stem diameter at 2.43 mm. The red LED application yielded the highest leaf count at 8.67 leaves, while the green LED application yielded the lowest leaf count at 6.43 leaves. While an increase in tuber formation was obtained in the red and full-spectrum LED applications, it is suggested that the green LED application did not form tubers due to the plant development being less than other LED applications.

Concerning second leaf stem length, second leaf width, and second leaf length, an examination of the leaf data from the first and second years showed that the best results for second leaf stem length and width in the white-flowered variety were 2.09 cm and 2.16 cm, respectively (see Table 3). No statistical difference was observed in second leaf length between the varieties. Furthermore, we also observed statistical difference in the examination of the fourth leaf stalk length, width, and leaf length data for the first and second years. Fourth leaf stalk length and width yielded the best results in the red-flowered variety at 1.6 cm and 1.99 cm, respectively, while fourth leaf length yielded the best results in the orange and white-flowered varieties at 3.93 cm and 4.02 cm, respectively.

Among the LED applications, the red LED application yielded the highest value of 2.46 cm for the second leaf stalk length, while the green LED application had the lowest value of 1.6 cm (see Table 4). For the second leaf width, the full spectrum (control) LED application yielded the highest value of 2.37 cm, while the green LED application yielded the lowest value of 1.64 cm. In the examination of second leaf length data, the full-spectrum and red LED applications yielded 4.81 cm and 5.03 cm, respectively, while the green LED application yielded 3.32 cm and the blue LED application yielded 4.51 cm. The full-spectrum application yielded the best results in second leaf data, while the lowest values were obtained in the green and blue LED applications. In LED applications, the best results for fourth leaf stalk length were obtained with red LED application at 1.95 cm, while for fourth leaf width and leaf length, the best results were obtained with full-spectrum and red LED application at 2.28 cm, 2.26 cm, and 4.34 cm, 4.50 cm, respectively (see Table 4).

The results of comparing the data for the first and second years revealed that the highest seedling height was 14.12 cm in the 30-day LED application, while the lowest seedling heights was 11.71 cm in the 15-day LED applications (see Figure 5). In our results, we observed no statistical difference in root length based on application durations. We also found that a 15-day application period resulted in 4.2 roots, while a 30-day application period resulted in 3.91 roots. We observed no statistical difference in stem diameter based on application durations. In terms of application duration, the 30-day application yielded the highest number of leaves at 8.37 leaves. Comparing the data for the first and second years revealed that the highest tuber formation was 0.24 in the 30-day LED application, while there was no tuber formation in the 15-day LED applications. LED application periods of 30 days yielded the best results in second leaf stem length, second leaf width, second leaf length and fourth leaf stem length, fourth leaf width, and fourth leaf length data.

## 4. Discussion

The results of comparing the data for the first and second years revealed that the highest seedling height was in the orange-flowered variety, while the lowest seedling heights were in the purple and red-flowered varieties. Ref. [14] reported that starflower seedlings with two to three true leaves and a height of 10–12 cm could be used for planting. Similar results were obtained in our study, and no negative effect of LEDs on seedling height was observed. Furthermore, several previous studies indicated that red LED applications promote seedling height [21,22,23,24]. In addition, it was reported that green LED applications do not affect seedling height in tomato and pepper seedlings, while blue LED applications have an inhibitory effect on seedling height [24,25,26]. There is also evidence to suggest that blue LED lamps inhibit plant growth by controlling plant height in ornamental plant cultivation [27]. In line with these findings, in the present study, we observed that blue LED applications have an inhibitory effect on seedling height in the second year. We observed that LED applications with 14 h of light and 10 h of darkness for 30 days had a greater effect on seedling height. Previously, the optimal photoperiod was reported to be 14 h [28,29].

Furthermore, the analysis of the root length data for the first and second years, as well as root number data, revealed that the red-flowered variety had the highest root length and root number. In LED applications, good results were obtained in root length except for the green LED application. In previous research, it was established that the use of control, red, and blue LED applications increases root activity by enhancing photosynthesis rates compared to daylight [30,31]. In our results, we observed no statistical difference in root length based on application durations. In terms of root number, the full-spectrum LED application yielded the best result, while the green LED application yielded the lowest value.

When examining the data for the first and second years in terms of body diameter, no statistical difference was observed between the varieties, while the full-spectrum LED application had the highest value. The green LED application yielded the thinnest stem diameter. Similarly, previous studies reported that white LED application increased stem diameter in pepper seedlings [29], while red LED application was found to increase stem diameter in tomato seedlings [24]. In the present study, we found that red LED application did not increase stem diameter as in tomato seedlings, while control LED application increased stem diameter as in pepper seedlings. Previous studies reported that the combined application of red and blue light in LED applications significantly increased the stem diameter of seedlings, with the lowest stem diameter obtained from red light application [32,33,34,35,36]. This trend can be attributed to photoreceptors (phytochromes) promoting cell division and development [35,37,38]. In our results, no statistical differences were found in stem diameter data based on application durations.

The red LED application yielded the highest leaf count, while the green LED application yielded the lowest leaf count. In terms of application duration, the 30-day application yielded the highest leaf count. In a previous study, the highest number of marketable leaves in lettuce was found in the blue + yellow + red LED light application, while the lowest number of marketable leaves was found in the red LED light application [39]. In the present study, we sought to determine the responses of LED applications between species, as red LED applications were found to increase the number of leaves. The decorative value of dahlias lies in their colorful flower clusters and leaves, which feature interesting petals of various colors, sizes, and shapes [40,41].

While an increase in tuber formation was obtained in the red and full-spectrum LED applications, it is suggested that the green LED application did not form tubers due to the plant development being less than other LED applications. Previous research documented that with an increase of temperature from 25 °C to 15 °C, tuberous root formation significantly increased [42]. In the present study, the plants were transferred from a climate chamber maintained at a constant temperature of 18–20 °C to a greenhouse with a temperature of 10–20 °C (day/night) and the data were collected after a 10-day acclimatization period. Based on these considerations, it can be concluded that the tubers formed during this process.

Concerning leaf petiole length, leaf width, and leaf length, an examination of the leaf data from the first and second years showed that the best results for leaf petiole length and width were found in the white-flowered variety. No statistical difference was observed in leaf length between the varieties. Among the LED applications, the red LED application yielded the highest value for the second leaf stalk length, while the green LED application had the lowest value. For the second leaf width, the full spectrum (control) LED application yielded the highest value, while the green LED application yielded the lowest value. These findings suggest that cultivation under light with 100% green wavelength has negative effects on plant growth [43,44]. In another study, the red LED application showed the most positive effect on mini red romaine lettuce, but the leaf shape was not normal [44]; such treatment resulted in leaf disorders, weak and abnormal plantlets in strawberry [45], and elongated but fragile stems in chrysanthemums [46]. In our study, leaves were observed to curl only in red LED applications (Figure 6). When the red light application was stopped, the leaves returned to their normal state. Ref. [47] reported that with red light at over 70%, petiole distortion was evident. Similarly, in this study, it was observed that 30 days of application period is sufficient for red LED application, but in longer periods of red LED application, seedlings cannot stand upright and tend to lodge.

Furthermore, we also observed statistical differences in the examination of the leaf stalk length, width, and leaf length data for the first and second years. Leaf stalk length and width yielded the best results in the red-flowered variety, while leaf length yielded the best results in the orange and white-flowered varieties. In LED applications, the best results for leaf stalk length were obtained with red LED application, while for leaf width and leaf length, the best results were obtained with full-spectrum and red LED application, over a 30-day application period. Previous studies documented that far-red light and full-spectrum LED applications significantly increased leaf area as compared to other wavelengths in tomato seedlings [34,48]. Furthermore, white light and full-spectrum LED applications also significantly increased leaf area [49,50]. Similarly, in the present study, full-spectrum and red LED applications yielded higher leaf data compared to blue and green LED applications.

## 5. Conclusions

Although this study appears preliminary compared with other studies, it uses four-star flower varieties and offers important guidance and can meaningfully contribute to the ornamental plant industry. The key findings were that red LED light increased seedling height, second leaf petiole length, fourth leaf petiole length, and leaf number, while full-spectrum LED light increased root number, stem diameter, and second leaf width. Furthermore, full-spectrum and red LED light yielded the highest values for second leaf length, fourth leaf width, fourth leaf length, and tuber formation. The analysis of our measurements and observations revealed seedling quality in the following varieties: red-flowered, white-flowered, orange-flowered and purple-flowered varieties. The use of LED light full-spectrum for 15 days had an insignificant effect on root number, stem diameter, root length, and second leaf length. For the other measurements and observations, the best results were obtained with 30 day full-spectrum and red LED light use. Furthermore, we find that it is possible to exploit the seedling height-limiting effect of blue LED applications. Excessive seedling growth is undesirable, which can be maintained with chemicals;paclobutrazol and uniconazole vd. Lastly, we do not find obstructions to the use of environmentally friendly LEDs.

Our results show the feasibility of using full-spectrum and red LED light for greenhouse plant production, and the viability of using these in combination lays the groundwork for future studies.

## Figures and Tables

**Figure 1 plants-14-02319-f001:**
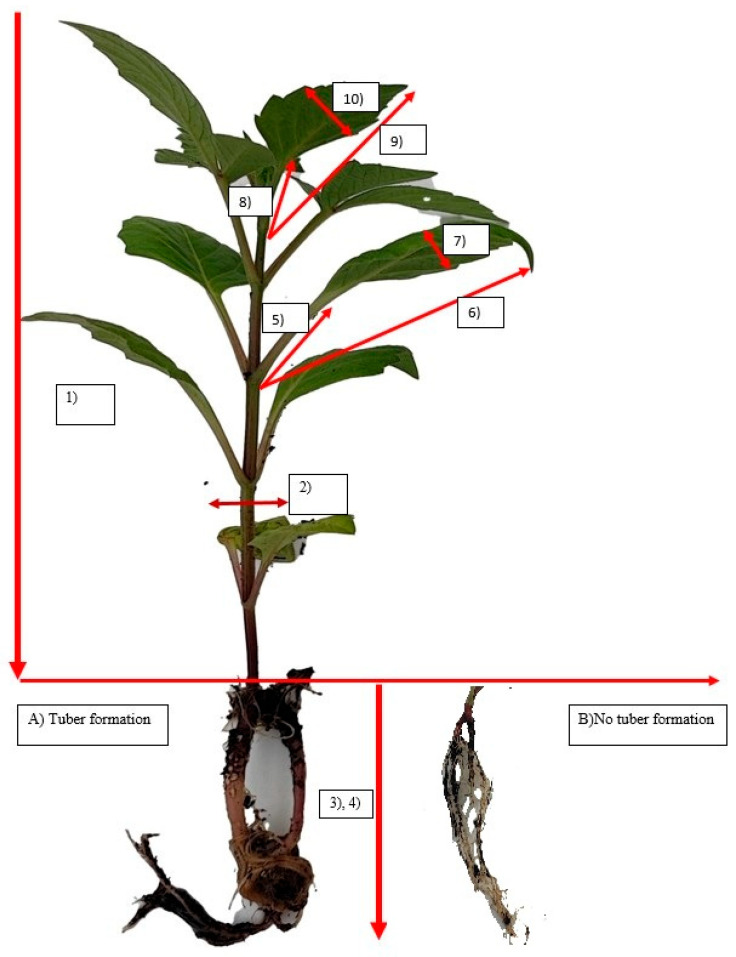
Dahlia Seedling: (A) Tuber formation (number): After the application, the roots of the seedlings were cleaned and the number of tubers was determined. (B) No tuber formation. (1) Seedling height (cm): The distance between the root collar and the tip of the shoot was taken as the seedling height. (2) Stem diameter (mm): Measured 1 cm above the cotyledon leaves. (3) Root length (cm): After uprooting, the longest root length developing on the seedling was measured (in cm). (4) Root number (number): After uprooting, the number of main roots developing on the seedlings was counted. (5) Second leaf stalk length (cm): Leaf stalk length was determined by measuring the stalks of the 2nd true leaves developing on the seedlings. (6) Second leaf length (cm): Leaf length was determined by measuring the 2nd true leaves developing on the seedlings. (7) Second leaf width (cm): The leaf width of the 2nd true leaves developing on the seedlings was measured. (8) Fourth leaf stalk length (cm): Leaf stalk length was determined by measuring the stalks of the 4th true leaves developing on the seedlings. (9) Fourth leaf length (cm): Leaf length was determined by measuring the 4th true leaves developing on the seedlings. (10) Fourth leaf width (cm): The leaf width of the 4th true leaves developing on the seedlings was measured.

**Figure 2 plants-14-02319-f002:**
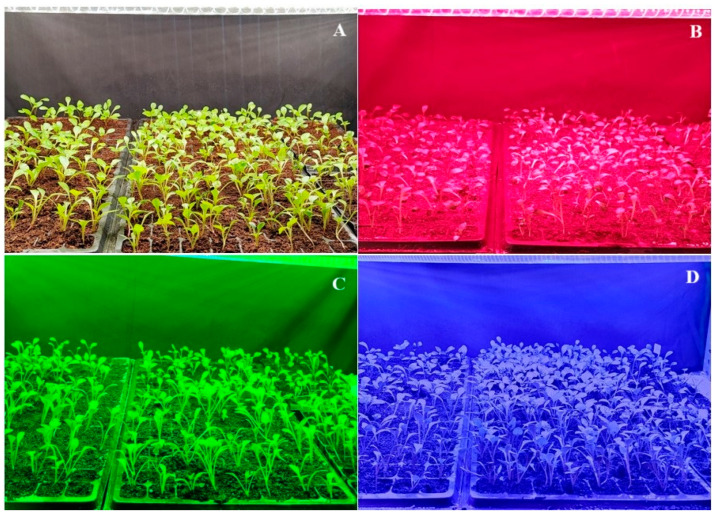
LED applications: (**A**) Full-spectrum LED application; (**B**) Red LED application; (**C**) Green LED application; (**D**) Blue LED application.

**Figure 3 plants-14-02319-f003:**
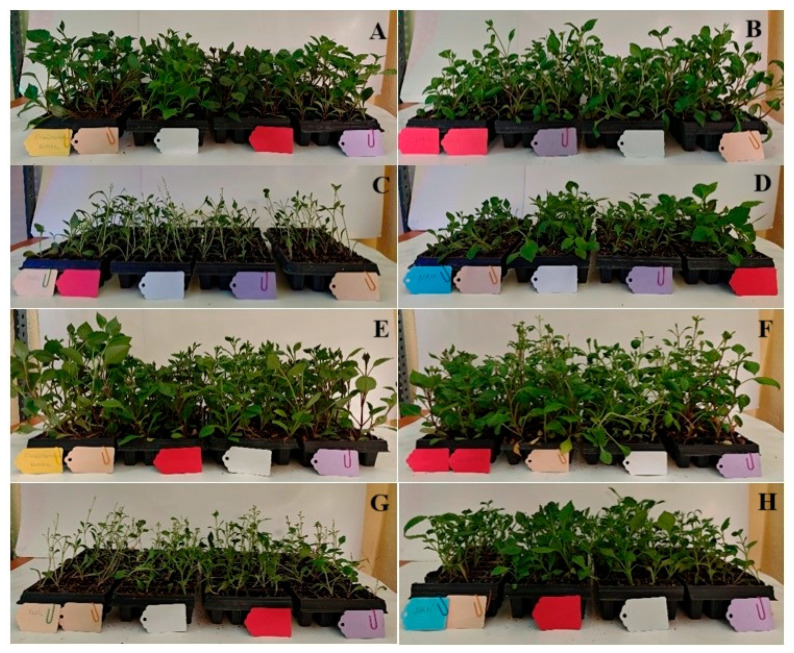
The seedlings obtained after the applications (15 days): (**A**) Full-spectrum LED application; (**B**) Red LED application; (**C**) Green LED application; (**D**) Blue LED application. The seedlings obtained after the applications (30 days): (**E**) Full-spectrum LED application; (**F**) Red LED application; (**G**) Green LED application; (**H**) Blue LED application.

**Figure 4 plants-14-02319-f004:**
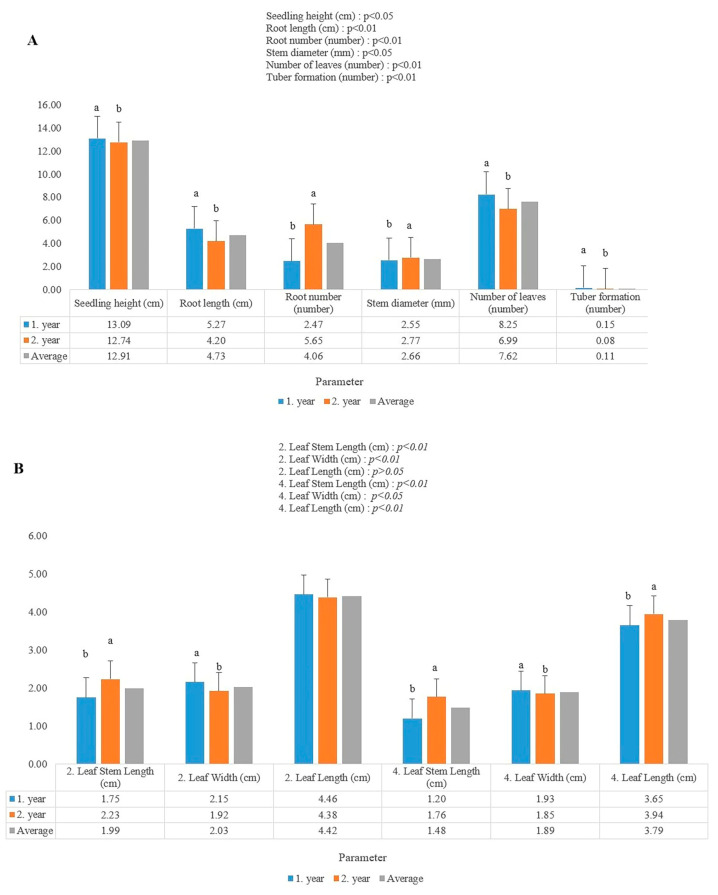
First year, second year and average: (**A**) Effects of the LED applications on the seedling height, root length, root number, stem diameter, number of leaves, tuber formation characteristics of the star flower (*Dahlia* sp.); (**B**) effects of the LED applications on the second leaf stem length, second leaf width, second leaf length, fourth leaf stem length, fourth leaf width, fourth leaf length characteristics of the star flower (*Dahlia* sp.). Different lowercase letters indicate least significant difference test; *p* < 0.05 and *p* < 0.01 indicate a significant difference; *p* > 0.05 = not significant.

**Figure 5 plants-14-02319-f005:**
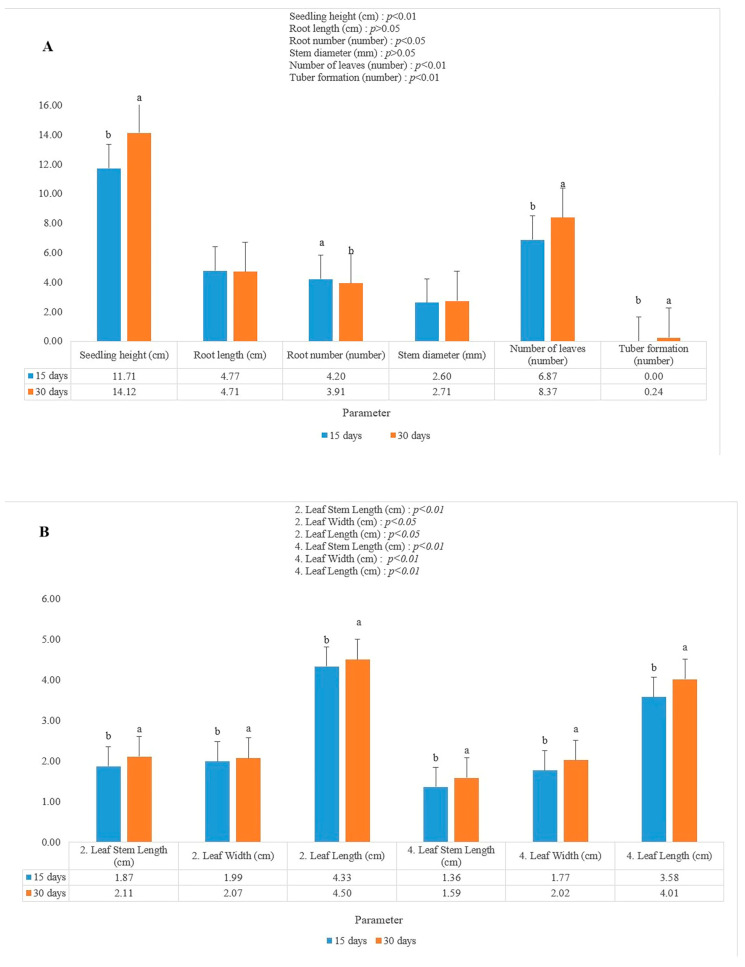
15 days, 30 days: (**A**) Effects of the LED applications on the seedling height, root length, root number, stem diameter, number of leaves, and tuber formation characteristics of the star flower (*Dahlia* sp.); (**B**) effects of the LED applications on the second leaf stem length, second leaf width, second leaf length, fourth leaf stem length, fourth leaf width, fourth leaf length characteristics of the star flower (*Dahlia* sp.). Different lowercase letters indicate least significant difference test; *p* < 0.05 and *p* < 0.01 indicate a significant difference; *p* > 0.05 = not significant.

**Figure 6 plants-14-02319-f006:**
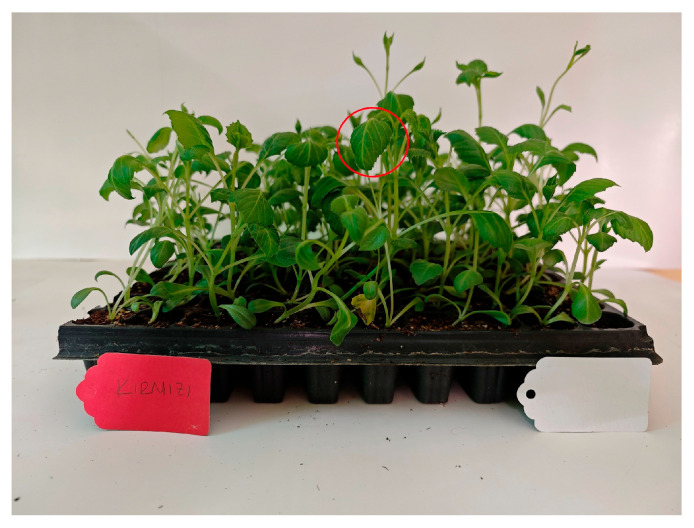
The leaf shape was not normal.

**Table 1 plants-14-02319-t001:** Effects of the first- and second-year applications on the variety characteristics of the star flower (*Dahlia* sp.).

	Variety					
Parameter	Violet	Orange	White	Red	LSD	f-Value
Seedling height (cm)	12.59 ± 0.12 c	13.80 ± 0.12 a	13.10 ± 0.12 b	12.17 ± 0.12 c	0.99	25.10 **
Root length (cm)	4.72 ± 0.59 ab	4.55 ± 0.59 b	4.78 ± 0.59 a	4.90 ± 0.59 a	2.74	4.62 *
Root number (number)	3.89 ± 0.07 b	3.92 ± 0.59 b	4.14 ± 0.59 ab	4.28 ± 0.59 a	0.3	4.12 *
Stem diameter (mm)	2.78 ± 0.07	2.61 ± 0.07	2.65 ± 0.07	2.59 ± 0.07	-	ns
Number of leaves (number)	7.37 ± 0.07 b	7.40 ± 0.07 b	7.75 ± 0.07 a	7.97 ± 0.07 a	5.64	10.21 **
Tuber formation (number)	0.14 ± 0.01 a	0.14 ± 0.01 a	0.13 ± 0.01 a	0.06 ± 0.01 b	0.11	5.21 **

Note. Small letters in the same column indicate a significant difference according to the LSD test. * and ** indicate a significant difference (*p* < 0.05 and *p* < 0.01, respectively). ns = not significant. LSD = least significant difference test.

**Table 2 plants-14-02319-t002:** Effects of the first- and second-year LED applications on the characteristics of the star flower (*Dahlia* sp.).

	LED Applications					
Parameter	Full-Spectrum	Red	Green	Blue	LSD	f-Value
Seedling height (cm)	14.82 ± 0.12 b	15.50 ± 0.12 a	10.82 ± 0.12 c	10.52 ± 0.12 c	0.99	348.18 **
Root length (cm)	4.98 ± 0.59 a	5.20 ± 0.59 a	3.70 ± 0.59 b	5.07 ± 0.59 a	3.77	102.52 **
Root number (number)	4.43 ± 0.07 a	4.13 ± 0.07 ab	3.58 ± 0.07 c	4.08 ± 0.07 b	0.41	15.05 **
Stem diameter (mm)	2.80 ± 0.07 a	2.67 ± 0.07 ab	2.43 ± 0.07 b	2.73 ± 0.07 ab	0.25	3.84 *
Number of leaves (number)	8.22 ± 0.07 b	8.67 ± 0.07 a	6.43 ± 0.07 d	7.16 ± 0.07 c	5.64	128.71 **
Tuber formation (number)	0.19 ± 0.01 a	0.22 ± 0.01 a	0.00 ± 0.01 c	0.06 ± 0.01 b	0.11	30.43 **

Note. Small letters in the same column indicate a significant difference according to the LSD test. * and ** indicate a significant difference (*p* < 0.05 and *p* < 0.01 respectively). ns = not significant. LSD = least significant difference test.

**Table 3 plants-14-02319-t003:** Effects of the first- and second-year applications on the characteristics of the star flower (*Dahlia* sp.).

	Variety					
Parameter	Violet	Orange	White	Red	LSD	f-Value
Second Leaf stem length (cm)	1.94 ± 0.36 b	1.99 ± 0.36 ab	2.09 ± 0.36 a	1.92 ± 0.36 b	1.64	3.34 *
Second Leaf width (cm)	2.09 ± 0.24 ab	1.87 ± 0.24 c	2.16 ± 0.24 a	2.01 ± 0.24 b	1.86	19.43 **
Second Leaf length (cm)	4.4 ± 0.73	4.33 ± 0.24	4.58 ± 0.24	4.36 ± 0.24	-	ns
Fourth Leaf stem length (cm)	1.32 ± 0.34 c	1.44 ± 0.34 bc	1.52 ± 0.34 ab	1.60 ± 0.34 a	2.09	9.54 **
Fourth Leaf width (cm)	1.88 ± 0.32 ab	1.84 ± 0.32 b	1.85 ± 0.32 b	1.99 ± 0.32 a	1.52	3.24 *
Fourth Leaf length (cm)	3.72 ± 0.46 b	3.93 ± 0.46 a	4.02 ± 0.46 a	3.52 ± 0.46 c	4.95	17.30 **

Note. Small letters in the same column indicate a significant difference according to the LSD test. * and ** indicate a significant difference (*p* < 0.05 and *p* < 0.01, respectively). ns = not significant. LSD = least significant difference test.

**Table 4 plants-14-02319-t004:** Effects of the first- and second-year LED applications on the characteristics of the star flower (*Dahlia* sp.).

	LED Applications					
Parameter	Full-Spectrum	Red	Green	Blue	LSD	f-Value
Second Leaf Stem Length (cm)	2.03 ± 0.36 b	2.46 ± 0.36 a	1.60 ± 0.36 d	1.86 ± 0.36 c	2.24	73.80 **
Second Leaf Width (cm)	2.37 ± 0.24 a	2.19 ± 0.24 b	1.64 ± 0.24 c	2.11 ± 0.24 b	1.86	200.73 **
Second Leaf Length (cm)	4.81 ± 0.73 a	5.03 ± 0.73 a	3.32 ± 0.73 c	4.51 ± 0.73 b	4.07	80.24 **
Fourth Leaf Stem Length (cm)	1.80 ± 0.34 b	1.95 ± 0.34 a	0.82 ± 0.34 d	1.34 ± 0.34 c	2.09	167.28 **
Fourth Leaf Width (cm)	2.28 ± 0.32 a	2.26 ± 0.32 a	1.16 ± 0.32 c	1.87 ± 0.32 b	2.09	192.85 **
Fourth Leaf Length (cm)	4.34 ± 0.46 a	4.5 ± 0.46 a	2.64 ± 0.46 c	3.72 ± 0.46 b	4.95	242.14 **

Note. Small letters in the same column indicate a significant difference according to the LSD test. ** indicates a significant difference (*p* < 0.01). ns = not significant. LSD = least significant difference test.

## Data Availability

The data related to the findings of this research are available upon request from the corresponding author.

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
