# Peer review of "Effects of LED Applications on Dahlia (Dahlia sp.) Seedling Quality"

_plants, 2025, doi:10.3390/plants14152319_

Round 1

Reviewer 1 Report

Comments and Suggestions for Authors

After reading this article, many questions and doubts arise.

  • Page 2, Line 65: I have not come across the exact date of sowing (month, day) being given in scientific articles. What does this information contribute to this article? What were the seeds soaked in?
  • Page 2, Line 68-69: Why was full-spectrum (control) irradiation initiated after dahlia seeds reached 50% germination? What was done with the seeds that did not germinate? Was any subsequent germination monitored?
  • Page 2, Line 72-73: Why were LEDs used in two time periods during dahlia cultivation: 15 and 30 days? Why weren't the selected LEDs with four different wavelengths used throughout the entire 1 and 2 year period of dahlia growth monitoring? Could the use of color LEDs for 15 and 30 days have had such a significant impact on metabolic changes in the dahlia seeds? Was the plant exposed to control light for the remaining cultivation period (1 or 2 years)? Was it unable to compensate for the metabolic changes during this time?
  • The methodology lacks information on whether the plants were watered during cultivation, and if so, how often. Were the plants fertilised, and were any plant protection products used during this time?
  • Page 2, Line 78-79: 'These applications were repeated over two years (two vegetation 78 periods).' - In my opinion, information on how to manage and monitor dahlia growth over a 2-year period should be described in more detail in the methodology section.
  • Figure 1 - Some of the numerical symbols mentioned in the description of image 1 are not highlighted, making it difficult to interpret. I propose describing some of the information in the methodology section (1, 2, 3, 4, 6, 11, 12).
  • This article lacks a 'Results' section. Instead, the authors have included a 'Discussion and  Conclusion' section.
  • The information in Table 1 is hard to read and needs to be improved - especially the 'day' column is incomprehensible.
  • How was tuber formation (%) determined?
  • The abstract of the article lacks information that the applications were repeated over two years (two growing seasons).

Author Response

Comments 1: Page 2, Line 65: I have not come across the exact date of sowing (month, day) being given in scientific articles. What does this information contribute to this article? What were the seeds soaked in?

Response 1: Page 2 Line 73: I have explained that we started using the LED light once there was germination. It should be clear that the seeds did not receive light at the beginning of the experiment.

Comments 2: Page 2, Line 68-69: Why was full-spectrum (control) irradiation initiated after dahlia seeds reached 50% germination? What was done with the seeds that did not germinate? Was any subsequent germination monitored?

Response 2: Page 2 Line 77-79: Germination occurred in the dark, and full-spectrum light was first applied to prevent rapid internode elongation (etiolation) of the germinated seeds. Later, when 50% of the second true leaves had developed, different LED wavelengths were applied. The failure of some plants to germinate did not affect the overall outcome of the experiment since we had enough replicates for the experiment.

Comments 3: Line 72-73: Why were LEDs used in two time periods during dahlia cultivation: 15 and 30 days? Why weren't the selected LEDs with four different wavelengths used throughout the entire 1 and 2 year period of dahlia growth monitoring? Could the use of color LEDs for 15 and 30 days have had such a significant impact on metabolic changes in the dahlia seeds? Was the plant exposed to control light for the remaining cultivation period (1 or 2 years)? Was it unable to compensate for the metabolic changes during this time?

Response 3: Page 2 Line 79-81: Preliminary studies have shown that longer light application times tend to cause seedlings to lodge and can lead to quality losses. Studies have also been reviewed that determine the effective application time. As there are not many experiments with Dahlia, our goal is to start with a single LED wavelength and continue with other wavelenght combinations. The seedlings from which data were  collected were not subsequently reused.

Comments 4: The methodology lacks information on whether the plants were watered during cultivation, and if so, how often. Were the plants fertilised, and were any plant protection products used during this time?

Response 4: Page 3 Line 90-92: The appropriate additions have been made to the article.

Comments 5: Page 2, Line 78-79: 'These applications were repeated over two years (two vegetation 78 periods).' - In my opinion, information on how to manage and monitor dahlia growth over a 2-year period should be described in more detail in the methodology section.

Response 5: Page 2-3, Line 73-95: Seedlings were planted, data were collected from the seedlings, and the seedlings were not reused thereafter. Seeds was procured again for the following season, and the process was repeated.

Comments 6: Figure 1 - Some of the numerical symbols mentioned in the description of image 1 are not highlighted, making it difficult to interpret. I propose describing some of the information in the methodology section (1, 2, 3, 4, 6, 11, 12).

Response 6: Page 4 Line 96-110: The appropriate additions have been made to the article.

Comments 7: This article lacks a 'Results' section. Instead, the authors have included a 'Discussion and  Conclusion' section.

Response 7: Page 11 Line 257: The appropriate additions have been made to the article.

Comments 8: The information in Table 1 is hard to read and needs to be improved - especially the 'day' column is incomprehensible.

Response 8: Removed from the article.

Comments 9: How was tuber formation (%) determined?

Response 9: Page 3 Line 97: The appropriate additions have been made to the article.

Comments 10: The abstract of the article lacks information that the applications were repeated over two years (two growing seasons).

Response 10: Page 1 Line 16: The appropriate additions have been made to the article abstract.

Reviewer 2 Report

Comments and Suggestions for Authors

This study reported an experiment involving four light qualities and two light cycles on four varieties of dahlias. At present, there are relatively few studies on the influence of light quality on dahlias. This research is innovative to some extent, but it still has some problems, as follows:

  1. Line 72, you can display the spectrograms of these four spectra and represent them with pictures. The full-spectrum (3,000 Kelvin) cannot show the specific spectra.
  2. line 76. Why is there a 10-day delay after the light treatment before measuring the data? And what are the light conditions during the 10-day period?
  3. line 116.Your experiment can be written as the first year and the second year. In Figure 2, I thought it was LED treatment for 1 year, which could be ambiguous.
  4. Table 1.Does it(1.) mean the first? Why not use "1st"? The same applies to other cases. It is recommended to make the changes. Because I have never seen such usage before.
  5. Table 1 can be placed in the Materials and Methods section to show the planting time.
  6. Table 2.The measured data does not have standard deviations. The data volume is small. It is recommended to measure some physiological indicators such as chlorophyll content.
  7. Line 134, this article did not conduct a comparison experiment on photoperiod. Why does it state that a 14-hour photoperiod is better?
  8. Line 217, 30 days is better only compared to 15 days. The specific number of days that is better requires further experiments.

Author Response

Comments 1: Line 72, you can display the spectrograms of these four spectra and represent them with pictures. The full-spectrum (3,000 Kelvin) cannot show the specific spectra.

Response 1: Page 4 Line 121: The appropriate additions have been made to the article.

Comments 2: Line 76. Why is there a 10-day delay after the light treatment before measuring the data? And what are the light conditions during the 10-day period?

Response 2: Page 3 Line 91-92: Instead of planting directly in the greenhouse under controlled conditions, seedlings were kept in an acclimation greenhouse for 10 days to minimize seedling loss. No additional light was applied to the seeds in the greenhouse.

Comments 3: Line 116.Your experiment can be written as the first year and the second year. In Figure 2, I thought it was LED treatment for 1 year, which could be ambiguous.

Response 3: Data for the 1st and 2nd years are given together.

Comments 4: Table 1.Does it(1.) mean the first? Why not use "1st"? The same applies to other cases. It is recommended to make the changes. Because I have never seen such usage before.

Response 4: The table has been removed from the article.

Comments 5: Table 1 can be placed in the Materials and Methods section to show the planting time.

Response 5: The clarifications have been removed from the article.

Comments 6: Table 2.The measured data does not have standard deviations. The data volume is small. It is recommended to measure some physiological indicators such as chlorophyll content.

Response 6: Page 6 Line 133, Page 7 Line 147-165, Page 8 Line 187, Page 10 Line 235: The tables have been updated.

Comments 7: Line 134, this article did not conduct a comparison experiment on photoperiod. Why does it state that a 14-hour photoperiod is better?

Response 7: Based on existing studies, we established the appropriate day lenght requirements of the plant.

Comments 8: Line 217, 30 days is better only compared to 15 days. The specific number of days that is better requires further experiments.

Response 8: Thank you, I have taken your suggestion into consideration.

Reviewer 3 Report

Comments and Suggestions for Authors

In this study GündoÄŸdu et al. decmonstrated that both red LED and full-spectrum light applications significantly improved the quality of dahlia (Dahlia sp.) seedlings, as indicated by greater seedling height, root length, root number, stem diameter, leaf dimensions, leaf number, and tuber formation—especially in red- and white-flowered varieties. The optimal results for root number were obtained after 15 days of LED exposure, while 30 days of red or full-spectrum LED application was most effective for promoting aboveground growth in most cultivars, including Figaro Orange. Blue LED light tended to inhibit seedling height. Therefore, red and full-spectrum LEDs for 30 days are recommended for producing high-quality dahlia seedlings, and both can be effectively used for commercial seedling cultivation of this crop. The study is intreseted however I have some suggestions:

  1. This is my suggestion, if possible, in current stage that morphological data were collected, direct physiological measurements (e.g., chlorophyll fluorescence, photosynthetic rate, or stomatal conductance) would provide deeper insight into how light spectra affect plant function.
  2. The study uses single-wavelength (100%) LED treatments for blue, red, and green, but plants typically benefit from mixed spectra. Future studies should include combinations of red and blue LEDs, as mixed wavelengths are often more effective for balanced growth.
  3. The varietal response suggests genotype-dependent effects. It is recommended to include more diverse genotypes and/or wild relatives to determine if findings generalize within the genus. Please add few sentences in discussion sentences of relvent litrature.
  4. The specifics of the "full-spectrum" treatment are not fully detailed. For clarity, there should be information on the actual spectrum or light intensity for each LED type. Please add in M&M
  5. If it is possible please change the table 2 in to graph form because the tables are dense and sometimes lack necessary units, error measurements (SD, SEM).

Author Response

Comments 1: This is my suggestion, if possible, in current stage that morphological data were collected, direct physiological measurements (e.g., chlorophyll fluorescence, photosynthetic rate, or stomatal conductance) would provide deeper insight into how light spectra affect plant function.

Response 1: Thank you, I will consider your points in future studies.

Comments 2: The study uses single-wavelength (100%) LED treatments for blue, red, and green, but plants typically benefit from mixed spectra. Future studies should include combinations of red and blue LEDs, as mixed wavelengths are often more effective for balanced growth.

Response 2: Page 13 Line 367: A single wavelength was preferred because there were data on different plants using wavelenght combinations, but no study exists for Dahlia.

Comments 3: The varietal response suggests genotype-dependent effects. It is recommended to include more diverse genotypes and/or wild relatives to determine if findings generalize within the genus. Please add few sentences in discussion sentences of relvent litrature.

Response 3: Page 12 Line 322-325: The appropriate additions have been made to the article.

Comments 4: The specifics of the "full-spectrum" treatment are not fully detailed. For clarity, there should be information on the actual spectrum or light intensity for each LED type. Please add in M&M

Response 4: Page 2 Line 83-88: The appropriate additions have been made to the article.

Comments 5: If it is possible please change the table 2 in to graph form because the tables are dense and sometimes lack necessary units, error measurements (SD, SEM).

Response 5: Page 6 Line 133, Page 7 Line 147-165, Page 8 Line 187, Page 9 Line 210, Page 10 Line 235: The appropriate additions have been made to the article.

Round 2

Reviewer 1 Report

Comments and Suggestions for Authors

Please standardize the font size in Figure captions.

Page 6 Line 130 - please check the number of appropriate table and figure

Author Response

Comments 1: Please standardize the font size in Figure captions.

Responce 1: Thank you, I have taken your suggestion into consideration.

Comments 2: Page 6 Line 130 - please check the number of appropriate table and figüre

Responce 2: Page 3 Line 88 (figüre 2), page 3 line 89 (figüre 3), page 6 line 142 (figüre 4), page 7 line 149 (figüre 4).

Reviewer 2 Report

Comments and Suggestions for Authors

The experimental design, data and paper structure have been improved after the author's revision. The following point can be checked again:

1.The bar values of tuber formation (number) in Figure 4A are all larger than the average, so you can check whether the data is wrong. Same as the Figure 5A. 

Author Response

Comments 1: The bar values of tuber formation (number) in Figure 4A are all larger than the average, so you can check whether the data is wrong. Same as the Figure 5A.

Response 1: Thank you, no errors were found in the data.